# Efficacy of Guided Tissue Regeneration Using Frozen Radiation-Sterilized Allogenic Bone Graft as Bone Replacement Graft Compared with Deproteinized Bovine Bone Mineral in the Treatment of Periodontal Intra-Bony Defects: Randomized Controlled Trial

**DOI:** 10.3390/jcm12041396

**Published:** 2023-02-09

**Authors:** Aniela Brodzikowska, Bartłomiej Górski, Marcin Szerszeń, Mariano Sanz

**Affiliations:** 1Department of Conservative Dentistry, Medical University of Warsaw, 02-097 Warsaw, Poland; 2Department of Periodontal and Oral Mucosa Diseases, Medical University of Warsaw, 02-097 Warsaw, Poland; 3Department of Prosthodontics, Medical University of Warsaw, 02-097 Warsaw, Poland; 4ETEP Research Group, Department of Dental Clinical Specialties, Faculty of Odontology, University Complutense of Madrid, 28040 Madrid, Spain

**Keywords:** intra-bony defect, periodontal regeneration, periodontitis, guided tissue regeneration (GTR), bone replacement graft, allograft, xenograft

## Abstract

(1) Background: The aim of this study was to compare the clinical and radiographic outcomes of guided tissue regeneration (GTR) using two biomaterials as bone replacement grafts in the treatment of periodontal intra-bony defects. (2) Methods: Using a split-mouth design, 30 periodontal intra-bony defects were treated with either frozen radiation-sterilized allogenic bone grafts (FRSABG tests) or deproteinized bovine bone mineral (DBBM, controls) combined with a bioabsorbable collagen membrane in 15 patients. Clinical attachment level gains (CAL-G), probing pocket depth reductions (PPD-R), and radiographic changes in linear defect fill (LDF) were evaluated 12 months postoperatively. (3) Results: The CAL, PPD, and LDF values improved significantly in both groups 12 months after the surgery. However, in the test group, the PPD-R and LDF values were significantly higher compared to the controls (PPD-R 4.66 mm versus 3.57 mm, *p* = 0.0429; LDF 5.22 mm versus 4.33, *p* = 0.0478, respectively). Regression analysis showed that baseline CAL was a significant predictor for PPD-R (*p* = 0.0434), while the baseline radiographic angle was a predictor for CAL-G (*p* = 0.0026) and LDF (*p* = 0.064). (4) Conclusions: Both replacement grafts when used for GTR with a bioabsorbable collagen membrane yielded successful clinical benefits in teeth with deep intra-bony defects 12 months postoperatively. The use of FRSABG significantly enhanced PPD reduction and LDF.

## 1. Introduction

Periodontitis is a chronic multifactorial inflammatory disease associated with dysbiotic plaque biofilms and is characterized by progressive destruction of the tooth-supporting apparatus. Its primary features include the loss of periodontal tissue support, manifested through clinical attachment loss (CAL) and radiographically assessed alveolar bone loss, occurrence of periodontal pocketing and gingival bleeding [1]. The presence of angular periodontal defects has been associated with increased severity, and if these lesions are not appropriately treated, there is a higher risk of further loss of periodontal support [2]. Moreover, residual probing depths (≥5 mm) combined with angular bony defects after periodontal therapy are independent predictors of progressive periodontal breakdown [3].

The goal of current periodontal therapy is, therefore, not only to arrest the progression of periodontitis, but also to reconstruct these angular bony lesions by periodontal regenerative surgical interventions. Among the most tested regenerative therapies, guided tissue regeneration (GTR) using barrier membranes, either alone or in combination with bone replacement grafts, has shown significant gains in CAL and bone compared with conventional periodontal therapy [4]. The use of the barrier membrane allows for epithelial exclusion and facilitates the re-population of the defect with those cells with differentiating capacity toward new periodontal attachment formation, while bone replacement grafts (either autografts, allogenic or xenogeneic bone substitutes) facilitate the maintenance of the space and stability of the blood clot in non-contained lesions. The selection of a specific graft biomaterial should be based on their inherent biological properties, demonstrate histological periodontal regeneration in pre-clinical research, and evidence of efficacy in randomized controlled clinical trials [5]. 

Deproteinized bovine bone mineral (DBBM) has been shown to possess characteristics facilitating periodontal regeneration such as high porosity to promote neovascularization, osteoconductivity to enhance growth and differentiation of periodontal tissue forming cells, and space maintenance capability to protect the blood clot and facilitate the wound healing process [6]. Similarly, allogenic bone replacement grafts have been regarded as a viable alternative to autografts and xenografts. Procured from human donors, allografts are available in various forms such as mineralized and demineralized bone matrix (DBM), demineralized freeze-dried bone allograft (DFDBA), and freeze-dried bone allograft (FDBA) [7]. Allogenic bone grafts have not only shown osteoconductivity, but also potential for osteoinduction by expressing bone morphogenetic proteins (BMPs-2, -4, -7) [8]. In fact, DFDBA has demonstrated histological evidence of new periodontal attachment formation [9]. However this potentiality depends on the donor (age, previous pathology, and drug therapy) and procurement and processing conditions of the graft material (sterilization methods, carrier use, preparation form) [7]. Although formulations are constantly emerging, the ideal allograft remains undefined. Frozen, radiation-sterilized, allogenic bone grafts (FRSABG) processed long bone of the lower extremities have been developed at the Department of Transplantology and Central Tissue Bank of the Medical University of Warsaw, and are available as bone chips, frozen at 70 °C, and subsequently sterilized by radiation at doses of 35 kGy in an accelerator with a high-speed electron beam [10,11]. Radiation-sterilized deep-frozen bone allografts have demonstrated improved osteoinductive properties and are remodeled faster compared to lyophilized irradiated bone grafts [12]. Although its safety and clinical efficacy in guided bone reconstruction of the maxilla and mandible have been evaluated, its efficacy in guided tissue regeneration when treating periodontal angular bony defects has never been verified [13]. 

The null hypothesis was based on the similar behavior of both bone grafts in terms of the improvement in the clinical and radiological parameters. It was, therefore, the purpose of this clinical trial to evaluate the efficacy of GTR using FRSABG as bone replacement grafts when compared with DBBM in the treatment of periodontal intra-bony defects. 

## 2. Materials and Methods

### 2.1. Study Design

The study was designed as a pilot split-mouth, double-blinded controlled clinical trial (RCT) in accordance with the Helsinki Declaration of 1975, as revised in Tokyo in 2013. The study design and protocol were approved by the University Bioethics Committee (KB/209/2017) and the study protocol was previously registered in ClinicalTrials.gov (Registration number: NCT03340012). Even though the surgeon was not blind to the treatment allocation, patients and researchers who evaluated the clinical and radiographic parameters were unaware of the allocation of defects to the tested interventions.

### 2.2. Study Sample

Participating subjects were recruited among periodontitis patients referred to the Department of Periodontology of the Medical University of Warsaw for periodontal treatment between June 2018 and December 2019 (Figure 1). Patients were thoroughly screened and were selected to participate in the RCT if they were 18 years or older, with a diagnosis of stage III or IV periodontitis, and with at least two teeth with probing pocket depth (PPD) of ≥6 mm, CAL of ≥6 mm, and the presence of an intra-bony angular lesion predominantly involving the interproximal area, identified on a digital periapical radiograph, with a defect depth (DD) ≥4 mm.

Patients were excluded if they were pregnant, smokers, had relevant systemic diseases, or under systemic medications. Selected teeth were excluded if exhibiting III-degree mobility, associated furcation involvement or with pulpal pathology, or with inadequate endodontic treatment.

Once selected, the patients were provided with oral hygiene instructions and non-surgical periodontal treatment. At re-evaluation, 3 months after, if demonstrating appropriate plaque and infection control, with full mouth plaque score (FMPS) ≤ 20% and full-mouth bleeding score (FMBS) ≤ 20%, and presence of at least two teeth with (PPD) of ≥6 mm, clinical attachment loss (CAL) of ≥6 mm, and an intra-bony angular lesion with a defect depth (DD) ≥ 4 mm, they were finally included in the RCT, once informed of the nature, potential risks, and benefits of participation in the study, and signing the informed consent form.

### 2.3. Surgical Intervention

Experimental and control regenerative surgeries were performed by a periodontist (BG) in a single surgical appointment. Depending on the width of the interdental space at the intra-bony defect site, either the modified papilla preservation technique (MPPT > 2 mm) or the simplified papilla preservation flap (SPPF when 2 mm or less) was used [14,15,16]. After applying the appropriate infiltrative local anesthesia, a full thickness mucoperiosteal flap was raised, exposing the intra-bony lesion, which was thoroughly debrided from and the affected teeth carefully root planned. Then, randomization and treatment allocation were carried out by opening sealed envelopes containing the assigned treatment modality. Randomization was prepared using a computer-generated randomization list by an independent investigator and was concealed throughout the study from the clinical and radiographic examiners. 

Once the experimental and control treatments were allocated, the surgeon filled the intra-bony lesion with either a frozen, radiation sterilized, allogenic, bone granules (FRSABG) in the test sites, or deproteinized bovine bone mineral granules (DBBM (Bio-Oss^®^, Geistlich Biomaterials, Princeton, NJ, USA) in the control sites. Patients were blind to the treatment mode. In both sites, a porcine-derived bioabsorbable collagen membrane (Bio-Gide^®^, Geistlich Biomaterials) was trimmed, closely filling the bone replacement graft and adapted to the anatomy of the defect. Flaps were then coronally repositioned without any tension (periosteum was released when necessary) to fully cover the regenerated site and secured with a combination of horizontal mattress sutures (Seralon 5/0 15 mm 3/8, Serag-Wiessner GmbH & Co., Naila, Germany) and a single vertical mattress suture positioned vertically in the inter-dental areas (Seralon 6/0 12 mm 3/8). Periodontal dressing was not used in any case.

Post-surgically, the patients were prescribed the use of anti-inflammatory medication (ibuprofen 600 mg twice a day for 2 days) and systemic antibiotic therapy (amoxicillin clavulanic acid (1 g) twice daily for 7 days). Patients also received postoperative instructions to avoid brushing or chewing for two weeks and to use an antiseptic rinse (0.2% chlorhexidine) for 3 weeks. Sutures were removed at 2 weeks when patients resumed careful brushing with a soft toothbrush. Patients were recalled every 2 weeks during the first 3 months postoperatively, and every 3 months for 1 year afterward.

### 2.4. Clinical Outcomes

Clinical parameters were assessed by a single experienced and calibrated examiner (MS). Calibration was carried out in five non-study stage III or IV periodontitis patients recording full-mouth PPD and CAL twice with an interval of 24 h and after achieving ≥90% of the recordings with an intra-examiner agreement within 1.0 mm in ≥90%. 

The primary outcome measurement was the clinical attachment level gains (CAL-gain) measured from the cemento–enamel junction (CEJ) to the base of the pocket, or from the most apical extension of the restoration/crown. These changes were registered before surgery and 12 months postoperatively at six sites in each affected tooth with a periodontal probe rounded off to the nearest millimeter, being a positive change indicative of CAL-gain (UNC probe 15 mm, Hu-Friedy, Chicago, IL, USA). The site with the greatest presurgical CAL value was used for the statistical analysis.

As secondary clinical outcomes, we evaluated the changes in probing pocket depth (PPD-R) and the changes in gingival recession (GR). Like with the CAL-gains, the site with the greatest presurgical CAL value was used for the statistical analysis. 

Full mouth plaque scores (FMPS) were the percentage of total surfaces with the presence of plaque [17] and full mouth bleeding scores (FMBS), assessed dichotomously, expressed as the percentage of pockets that bled after gentle probing [18].

Intra-surgical measurements included: Defect depth as the distance between the bottom of the defect and the most coronal point of the bony walls surrounding the defect;Defect width as the distance from the most coronal point of the bony walls surrounding the defect to the root surface;Defects were classified as one-wall, two-wall, and three-wall defects depending on the number of remaining walls.

### 2.5. Radiological Outcomes

Radiographic changes in the treated intra-bony lesions were measured from standardized periapical radiographs taken before the surgery and at 12 months postoperatively. Periapical radiographs were taken using the parallel long cone technique using a customized film-holder device and a bite block. These radiographs were digitized and measured with image analysis software by selecting the following landmarks: the CEJ, the alveolar crest (AC), and base of the defect (BD) [19]. Then, a primary line was drawn through the tooth axis (AUX1) and a secondary line from AC (AUX2) was drawn perpendicular to AUX1 (Figure 2). The distance from the point where AUX2 crossed the CEJ–BD line to the base of the defect was considered as the defect depth (DD). Linear defect fill (LDF) was calculated by subtracting the CEJ–BD distance at the end of 12 months from CEJ–BD at the baseline, while the percentage defect fill (%DF) was obtained by dividing the LDF by DD at the baseline [20]. Radiographic DD, LDF, and %DF were considered the secondary outcome variables. The radiographic defect angle was calculated between the AUX1 and AUX2 lines of the involved tooth [21].

All radiographic measurements were performed by an experienced and calibrated clinician who was blinded with respect to the surgical intervention. An intra-examiner calibration exercise, similar to the clinical calibration, was performed by examining 10 non-study-related radiographs prior to the commencement of the study.

### 2.6. Patient-Reported Outcome Measures (PROMS)

Patient perception of post-operative morbidity and treatment satisfaction were evaluated using a questionnaire based on visual analogue scales (VAS), which was completed 2 weeks after surgery. Morbidity measured discomfort, pain, edema, interference with daily activities, and eating and speech impairments [22]. Additionally, patients were asked to write down the number of anti-inflammatory medication tablets taken beyond the first two pills. 

At the 1-year postoperative visit, the patient’s treatment satisfaction was evaluated by asking: (1) ‘Overall, how satisfied are you now with the results of the surgery?’; (2) ‘If you had to make the decision again, how likely would you be to accept the same surgery?’ [23]. 

### 2.7. Data Analysis

The null hypothesis was based on similar performance in the primary outcome (CAL-G at 1-year post-surgery) for the tested (FRSABG) bone replacement graft compared to the standard of care (DDBM). The sample size was set a priori at 15 patients (30 defects per arm) based on the pilot nature of this study, since the examined biomaterial had not been previously tested in a RCT.

Outcome variables were tested for their distribution normality using the Shapiro–Wilk test and were described as the means, standard deviations (SD), and 95% confidence intervals (CI). Intergroup comparisons at 1-year were performed using the parametric test *t* test for independent variables, while the intragroup changes (baseline-1-year) were evaluated by the *t* test for paired data. PROMs were also evaluated with the *t* test for independent samples. Multiple linear regression was used to assess the relationship of age, sex, tooth type, surgical procedure (FRSABG versus DBBM), FMPS, FMBS, PPD, CAL, DD, and RVG angle at 1 year (independent variables) with CAL gain, PPD reduction, and LDF after 12 months. 

All statistical analyses were performed using dedicated software (Statistica^®^ 13 data analysis) and any *p* values less than 0.05 were considered statistically significant. 

## 3. Results

### 3.1. Baseline Clinical Characteristics

Fifteen patients (nine women and six men) with a mean age of 38.7 ± 7.6 years were enrolled in this RCT. All subjects completed the 12-month follow-up. One patient had one affected tooth extracted due to a root fracture (test site). Consequently, 14 teeth in the test group and 15 teeth in the control were suitable for analysis at the 1-year post-operatory visit. 

At the baseline, the patients demonstrated a good level of oral hygiene and appropriate degree of infection control as demonstrated by FMPS: 8.45% ± 7.33 and FMBS: 13.24% ± 5.62, respectively. Baseline characteristics of the treated sites are depicted in Table 1, demonstrating a well-balanced distribution between the treatment arms. Complete gingival wound closure was accomplished for all defect sites. However, membrane exposure was observed at 2 weeks after surgery in three of the test sites and three of the control sites. Exposed areas were rinsed with a 0.2% chlorhexidine solution and 1% chlorhexidine gel was applied topically daily until complete re-epithelialization was achieved. 

### 3.2. Changes in Clinical and Radiological Parameters

Table 2 depicts the values of the clinical and radiographical outcome variables. In both groups, the intra-group differences were statistically significant, thus demonstrating the positive performance of both treatment approaches regarding the CAL, PPD, and GR values. The primary outcome measurement (CAL-G at 1-year) showed higher values in the test group (FRSABG), although differences in the standard of care (DDBM) were not statistically significant (5.54 mm (C.I. 4.40–6.55) versus 4.54 mm (C.I. 3.95–5.22), respectively, *p* = 0.0891). However, differences in PPD-R were significantly higher in the test versus the control (4.66 mm (C.I. 3.78–5.51) vs. 3.57 (C.I. 3.18–5.22), respectively, *p* = 0.0190). Differences in recession increase (GR) were also similar between the test and control groups (0.22 mm (C.I. −0.28–0.75) versus 0.45 (C.I. 0.11–0.78), respectively, *p* = 0.3281). Conversely, differences in radiographic LDF were significantly higher in the test versus the control (5.22 (C.I. 4.56–5.54) versus 4.33 (C.I. 3.40–5.18), respectively, *p* = 0.0478), resulting in a %DF of 85.89% and 83.27% for the test and control groups, respectively (Figure 3).

### 3.3. Regression Analysis

Table 3 depicts the results of the regression analysis using CAL-G, PPD-R, and LDF at 1 year postoperatively as the dependent variables. The data showed no multi-collinearity and for the main outcome measurement, 56% of the variability could be explained by the regression model (R^2^ = 0.5649). However, only the RVG angle was a significant predictor for CAL-G (*p* = 0.0026), and each increase in the RVG angle resulted in 0.16 mm of lesser CAL-G. Baseline CAL was a significant predictor for PPD-R (*p* = 0.0434), and each 1 mm increase in baseline CAL led to a higher PPD-R (0.42 mm). Similarly, the RVG angle was the only significant predictor of LDF (*p* = 0.0064). The increase in RVG of 1 degree decreased LDF by 0.11 mm. More than 70% of the variability could be explained by the regression model (R^2^ = 0.7145). Surgical technique (FRSABG versus DBBM) was not associated with better outcomes (*p* > 0.05).

### 3.4. Patient-Reported Outcomes

Table 4 depicts the post-operative morbidity. Most patients reported discomfort, pain, and edema of mild intensity, without significant differences between the treatment modalities, with the mean additional painkiller intake of 2.17 ± 0.75 tablets. One year after treatment, all patients were satisfied with the results of the surgery (test sites: VAS 92.14 ± 8.07; control sites 91.14 ± 8.40, with 0 = no satisfaction and 100 = maximum satisfaction), and they stated that they would make the same decision regarding treatment if necessary (VAS 92.00 ± 8.64 and 90.38 ± 8.33, respectively).

## 4. Discussion

Results of the present RCT demonstrated significant improvements in the clinical and radiographic outcomes, 12 months post-surgically, for both periodontal regenerative treatment modalities (FRSABG + MEM and DBBM + MEM). The results in the main outcome (CAL-G) showed 5.54 mm and 4.54 mm gains for the test and control sites, respectively, although differences between treatments were not statistically significant (1.00 mm difference, *p* = 0.0891). Similarly, differences in the increase in GR were similar when comparing the test and control sites. The use of FRSABG + MEM, however, resulted in significantly higher PPD reductions and enhanced the LDF (PPD-R 4.66 mm versus 3.57 mm, 1.09 mm difference (*p* = 0.0478), LDF 5.22 mm versus 4.33, 0.89 mm difference *p* = 0.0429) when compared with DBBM + MEM. These results are congruent with a recent systematic review and meta-analysis evaluating GTR interventions with significant CAL gains (1.15 mm) and PPD reductions (1.24 mm) compared with flap surgery alone, although differences in gingival recession and radiographic bone gain were not statistically significant [4]. When compared with similar studies using bone allografts within the GTR procedures, Majzoub et al. [24] reported 3.55 ± 1.85 mm CAL-gains and 3.87 ± 1.87 mm PPD-reductions 1-year after GTR treatment with freeze-dried bone allograft or solvent-dehydrated bone allograft. Kher et al. [25] also evaluated the effect of GTR combining a human allograft with a collagen membrane and reported a mean CAL-gain of 3.54 ± 0.36 mm and a mean PPD-reduction of 4.06 ± 0.38 mm 12 months after the surgery. Moreover, bone fills ranging from 1.3 to 2.6 mm and from 1.7 to 2.9 mm have been reported in RCTs using FDBA and DFDBA, respectively [26,27]. A recent study combining DFDBA and platelet-rich fibrin (PRF) reported CAL gains of 3.40 ± 1.65, PPD reductions of 3.60 ± 1.42 mm, and a radiographic DD reduction of 5.35 ± 2.05 mm), these results being significantly improved compared to using open flap debridement and DFDBA only [28]. Compared with the present RCT, these studies report either similar or inferior outcomes, which may be explained by the patient selection methods used and the high osteoinductive potential of FRSABG. Moreover, the presence of collagen within the graft could support the initial mineral deposition and the organization of crystal growth. The sterilization dose of 35 kGy is high enough to deactivate potential pathogens contaminating a graft, while being low enough to preserve the optimal biological and mechanical tissue properties [29]. This graft processing method may preserve the osteoconductive potential of rhBMPs and thus potentiate the guided tissue effect when combined with the collagen barrier. It was observed that allogenic deep-frozen bone matrices irradiated with doses of 35 kGY at −72 °C induced de novo bone formation after transplantation [11]. 

In the present study, we used DBBM as a control bone replacement graft due to its well-demonstrated ability to trigger periodontal regeneration in both preclinical and clinical studies [4]. In fact, favorable clinical and radiographic results were reported in this RCT in the control group, with significantly higher CAL gain (4.54 mm) and PPD reduction (3.57 mm) values, which are comparable with previous reports using the same biomaterial. Sculean et al. [30] reported an average CAL gain of 4.0 ± 1.3 mm and an average PPD reduction of 5.3 ± 1.6 mm at 1 year using DBBM in combination with a collagen membrane. However, the application of FRSAB was associated with greater CAL gains and significantly higher PPD reductions and LDF, which can be explained by the potential enhanced osteoinduction of FRSAB. Greater PPD reduction may be of clinical relevance, since residual deep pockets after periodontal therapy have been associated with an increased risk for disease progression and tooth loss [31]. 

Patient-related factors, defect morphology, and the use of enhanced surgical techniques have been reported as critical factors influencing the efficacy and predictability of GTR procedures in periodontal regeneration interventions [8]. In this study, patient-related variables were rigorously controlled and may have had a relevant impact on the dimension of the changes reported in this RCT. Results from the multiple linear regression analysis clearly showed that the lesion and its morphology influenced the reported results. In fact, the baseline CAL was significantly related to PPD-R, whereas the baseline radiographic angle significantly affected CAL-G and LDF. 

Previous studies have also reported the number of remaining bony walls as a predictor of CAL gain, and the radiographic defect depth and angle as influencing the changes in bone density 12 months after GTR using DBBM [32]. However, in a large multicenter RCT using this biomaterial, the presurgical defect angle did not significantly influence the clinical outcomes after GTR [33]. When compared with RCTs using allogenic grafts as bone replacement grafts in GTR procedures, Majzoub et al. [24] reported that smoking [−0.91 (95% CI: −1.73, −0.07), *p* = 0.03] and membrane exposure [−1.18 (95% CI: −2.28, −0.06), *p* = 0.03] were the main factors associated with lower CAL-G, while the initial PPD correlated to higher CAL-G [0.57 (95% CI: 0.16, 0.97), *p* = 0.006] at 1 year post-surgery. 

This RCT also evaluated the impact of the tested GTR interventions on different aspects of the patients’ quality of life, with most patients reporting some discomfort, pain, edema, and eating impairment after the surgical intervention, but without any significant differences when comparing the tested treatment modalities. Furthermore, all patients were satisfied with the result of the surgery (test sites: VAS 92.14 ± 8.07; control sites 91.14 ± 8.40) and they responded positively to the decision to again undertake a similar surgical intervention if necessary. This proves that the discomfort reported during the early post-operative period had no impact on overall treatment satisfaction. Although few studies have evaluated patient-reported outcomes associated with GTR [34], the available data showed that patient-reported outcome measures were very similar to those observed in the current study [35]. These positive findings could be explained by the careful surgical planning used in this RCT, where papilla preservation flaps were used depending on the available interdental space and the fact that membrane exposure during postoperative healing was an infrequent event (20%). This incidence compares well with data from a systematic review on GTR reporting an incidence ranging between 20% and 86% [36].

The results from this RCT, however, should be evaluated with caution in light of the limitations of this clinical research. Due to the absence of previous clinical studies using the tested biomaterial, we were unable to properly calculate the sample size, which resulted in designing this split-mouth RCT as a pilot study, thus establishing an arbitrary sample population. There exists, therefore, a need for further studies to properly evaluate the efficacy of this promising biomaterial, most likely using the post-hoc results from this trial to effectively calculate the required sample population. It is also likely that, due to the relatively small number of subjects in this RCT, some variables and predictors might not have displayed significant associations in the multivariate analysis, and due to multi-collinearity, the results of the subgroup analyses should be interpreted with caution. Furthermore, the radiographic outcomes achieved in both the test and control sites should be interpreted with caution, since both biomaterials have a slow resorption rate and consequently, the observed radiographic changes may relate to the presence of the biomaterial inside the defect, rather than the actual formation of new bone. Finally, as is usual in clinical studies evaluating the efficacy of periodontal regenerative therapies, we were unable to assess the nature of healing since we did not evaluate the histological outcomes. 

## 5. Conclusions

Within the limitations of this RCT, it can be concluded that the GTR of periodontal intra-bony defects using frozen non-decalcified radiation-sterilized allogenic bone grafts as bone replacement grafts under a bioabsorbable collagen membrane resulted in significant improvements in the clinical and radiographic parameters over a 12-month period, similar to those achieved with GTR using the gold standard biomaterial as a bone replacement graft (DBBM). However, the application of the tested allogenic graft additionally enhanced the probing pocket depth reduction and radiographically observed the linear defect fill. It can thus be concluded that the FRSAB graft may represent a potential new material for the treatment of periodontal intra-bony defects when combined with a bioabsorbable collagen membrane. With that in mind, well-designed future clinical trials with longer follow-up and larger sample sizes are required to further evaluate and confirm the efficacy of FRSABG for periodontal regeneration and the long-term stability of the obtained clinical outcomes.

## Figures and Tables

**Figure 1 jcm-12-01396-f001:**
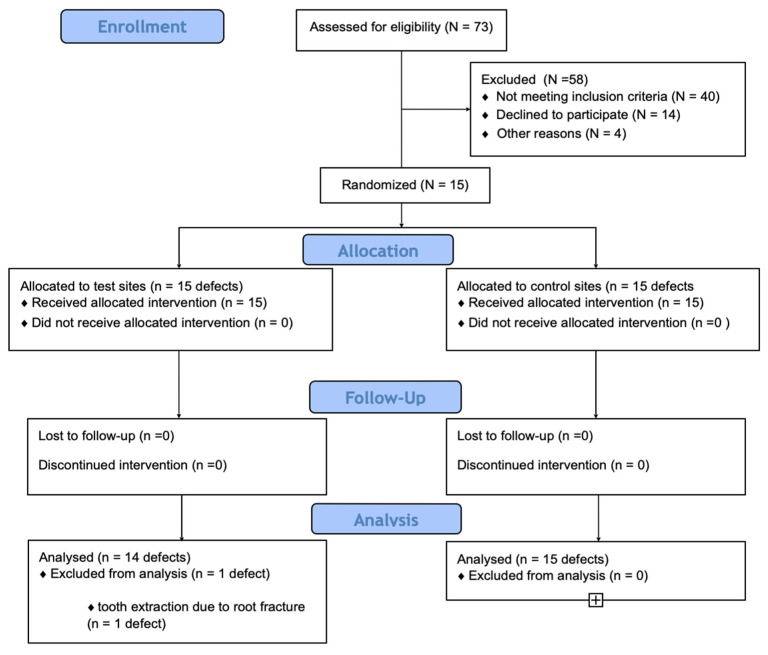
Consort diagram showing the study outline.

**Figure 2 jcm-12-01396-f002:**
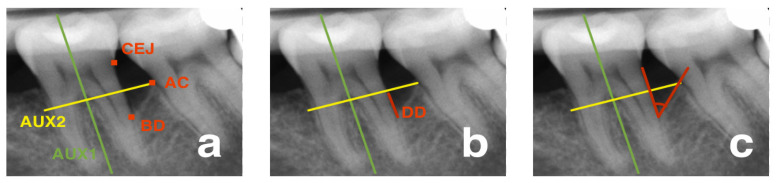
Radiographic measurement of the intra-bony defect. (**a**) AUX1—auxiliary line 1, AUX2—auxiliary line 2, CEJ—cemento–enamel junction, AC—alveoral crest, BD—base of the defect. (**b**) DD—defect depth. (**c**) Defect angle.

**Figure 3 jcm-12-01396-f003:**
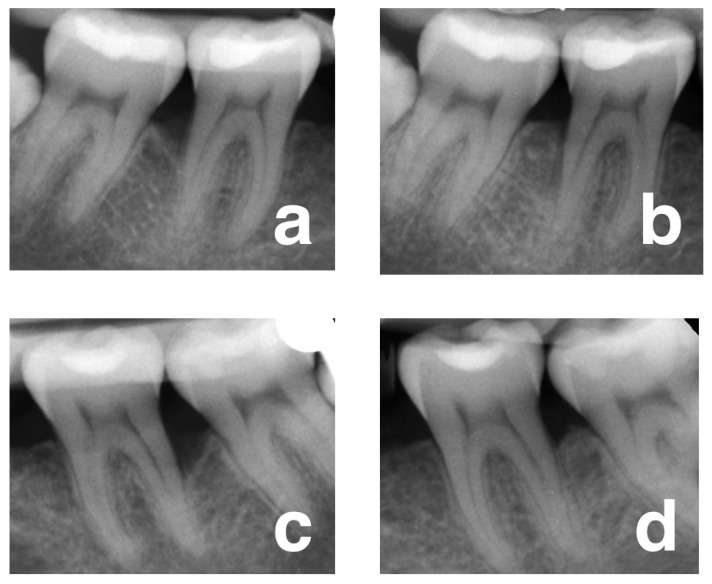
(**a**) Baseline radiograph of the intra-bony defect on the distal surface of tooth 46 (test site). (**b**) Radiograph at 12 months post-surgery. (**c**) Baseline radiograph of intra-bony defect on the distal surface of tooth 36 (control site). (**d**) Radiograph at 12 months post-surgery.

**Table 1 jcm-12-01396-t001:** Baseline clinical characteristics: tooth type, tooth position, radiographic angle (RVG angle), intra-surgical measurements, and defect morphology [mean, 95% confidence interval (CI), standard deviation (SD)].

Variables	Test Sites (*n* = 15)	Control Sites (*n* = 15)	*p*
Tooth type (*n*)			
Molars	6	6	
Premolars	5	5	
Upper incisors, canines	4	4	
Tooth position (*n*)			
Maxillary teeth	7	8	
Mandibular teeth	8	7	
Radiographic angle (degrees)	23.39 (20.39–26.40) ± 5.42	26.49 (22.04–30.45) ± 7.59	0.2467
Intra-surgical measurements (mm)			0.8029
Defect depth	6.00 (4.99–7.00) ± 1.81	5.80 (4.42–7.18) ± 2.48	
Defect width	3.73 (2.97–4.50) ± 1.39	3.06 (2.53–3.60) ± 0.96	0.1372
Defect morphology (*n*)			
One-wall	5	4	
Two-wall	5	6	
Three-wall	5	5	

*n* = number of split-mouth defects.

**Table 2 jcm-12-01396-t002:** Baseline clinical characteristics: tooth type, tooth position, radiographic angle (RVG angle), intra-surgical measurements and defect morphology [mean, 95% confidence interval (CI), standard deviation (SD)].

	Baseline	12 Months	*p* (Baseline-1-Year)	∆ Change (Baseline-1-Year)
PPD test (mm)	7.67 (6.98–8.35) ± 1.23	3.23 (2.69–3.65) ± 0.83	<0.0001 * <0.0001 *	4.66 (3.78–5.51) ± 1.24
PPD control	7.40 (6.82–9.66) ± 1.06	3.63 (3.21–4.06) ± 0.77		3.57 (3.18–5.22) ± 1.11
*p* (test vs. control)	0.3427	0.0429 *		0.0190 *
CAL test (mm)	8.93 (8.13–9.73) ± 1.44	3.23 (2.69–3.65) ± 0.83	<0.0001 * <0.0001 *	5.54 (4.40–6.55) ± 1.18
CAL control	8.73 (7.81–9.69) ± 1.67	4.13 (3.25–5.02) ± 1.59		4.54 (3.95–5.22) ± 1.11
*p* (test vs. control)	0.2219	0.1182		0.0891
GR test (mm)	1.21 (5.21–6.57) ± 1.23	0.66 (0.42–0.91) ± 0.45	0.4981 0.2610	0.22 (−0.28–0.75) 0.91
GR control	1.47 (0.84–2.08) ± 1.12	1.03 (0.52–1.55) ± 0.93		0.45 (0.11–0.78) ± 0.66
*p* (test vs. control)	0.7918	0.1651		0.3281
DD test (mm)	5.89 (5.21–6.57) ± 1.23	0.66 (0.42–0.91) ± 0.45	<0.0001 * <0.0001 *	-
DD control	5.32 (4.31–6.35) ± 1.84	0.92 (0.58–1.27) ± 0.61		
*p* (test vs. control)	0.2811	0.0334 *		
LDF test (mm)	-	5.22 (4.56–5.54) ± 1.11	-	-
LDF control		4.33 (3.40–5.18) ± 1.74		
*p* (test vs. control)		0.0478 *		
%DF test (mm)	-	85.89 (81.29–95.11) ± 8.90	-	-
%DF control		83.27 (77.61–90.37) ± 11.41		
*p* (test vs. control)		0.1091		

PPD—probing pocket depth, CAL—clinical attachment level, GR—gingival recession, DD—radiographic defect depth, LDF—linear defect fill, %DF—percentage defect fill, * Statistically significant (*p* ≤ 0.05).

**Table 3 jcm-12-01396-t003:** Regression analysis with the clinical attachment level (CAL) gain (mm), probing pocket depth (PPD) reduction (mm), and (LDF) linear defect fill (mm).

Parameter	Regression Coefficient	Standard Error	Confidence Interval		*p*
			Lower	Upper	
CAL gain from baseline to 1 year as dependent variable. *R*^2^ = 0.5649					
Intercept	5.6065	3.5427	7.0054	4.2077	0.1319
Gender	0.3713	0.6305	0.6202	0.1223	0.5636
Age	−0.0308	0.0570	−0.008	−0.0534	0.5955
Surgical procedure (test vs. control)	−0.1939	0.5223	0.0123	−0.4001	0.7150
Tooth type (incisors, canines, premolars vs. molars)	−0.3964	0.3969	−0.239	−0.5532	0.3319
Tooth position (upper vs. lower)	1.0250	0.5503	1.2423	0.8077	0.0798
FMPS	−0.0826	0.0581	−0.059	−0.1056	0.1735
FMBS	0.0778	0.0456	0.0959	0.0598	0.1062
PPD	−0.1706	0.3491	−0.0328	−0.3085	0.6312
DD	−0.0306	0.2670	0.0748	−0.1360	0.9100
RVG angle	−0.1669	0.0474	−0.1482	−0.1856	0.0026 *
PPD reduction from baseline to 1 year as dependent variable. *R*^2^ = 0.6335					
Intercept	4.2998	2.4676	5.2742	3.3255	0.0984
Gender	−0.3789	0.5490	−0.162	−0.5957	0.4988
Age	−0.0848	0.0486	−0.065	−0.1041	0.0982
Surgical procedure (test vs. control)	0.1001	0.4756	0.2879	−0.0877	0.8356
Tooth type (incisors, canines, premolars vs. molars)	−0.3512	0.3612	−0.2085	−0.4939	0.3437
Tooth position (upper vs. lower)	0.4019	0.5002	0.5994	0.2044	0.4321
FMPS	0.0029	0.0527	0.0237	−0.0179	0.9566
FMBS	0.0662	0.0414	0.0826	0.0498	0.1274
CAL	0.4270	0.1965	0.5046	0.3494	0.0434 *
DD	0.1468	0.2418	0.2423	0.0513	0.5514
RVG angle	−0.0771	0.0407	−0.061	−0.0932	0.074
LDF from baseline to 1 year as the dependent variable. *R*^2^ = 0.7145					
Intercept	6.3908	3.0015	7.5760	5.2057	0.0473
Gender	−0.1927	0.5437	0.0219	−0.4074	0.7270
Age	−0.0066	0.0504	0.0133	−0.0265	0.8969
Surgical procedure (test vs. control)	0.5582	0.4464	0.7345	0.3820	0.2270
Tooth type (incisors, canines, premolars vs. molars)	−0.6148	0.3317	−0.4838	−0.7459	0.0803
Tooth position (upper vs. lower)	−0.0079	0.4854	0.1837	−0.1996	0.9871
FMPS	0.0125	0.0504	0.0324	−0.0073	0.8066
FMBS	0.0000	0.0392	0.0155	−0.0154	0.9992
PPD	−0.1305	0.3070	−0.0093	−0.2518	0.6756
CAL	0.3715	0.2162	0.4569	0.2861	0.1029
RVG angle	−0.1191	0.0386	−0.1038	−0.1343	0.0064 *

FMPS—full-mouth plaque score, FMBS—full-mouth bleeding score, PPD—probing pocket depth, DD—radiographic defect depth, CAL—clinical attachment level, * Statistically significant (*p* ≤ 0.05).

**Table 4 jcm-12-01396-t004:** Subject experience in terms of post-operative morbidity (*N* = 15).

	Test (*n* = 15)	Control (*n* = 15)		Test (*n* = 15)		Control (*n* = 15)		
	Number of Subjects (%)	Number of Subjects (%)	*p* Value	Intensity (VAS) Mean ±SD	Minimum–Maximum	Intensity (VAS) Mean ±SD	Minimum–Maximum	*p* Value
Discomfort	11 (73.33)	12 (80.00)	1	27.27 ± 21.33	2–57	28.42 ± 18.64	3–55	0.8921
Pain	11 (73.33)	12 (80.00)	1	28.73 ± 15.95	2–45	27.08 ± 14.13	3–45	0.7957
Edema	11 (73.33)	12 (80.00)	1	23.40 ± 14.45	3–47	33.09 ± 18.16	3–57	0.1949
Eating impairment	14 (93.33)	15 (100)	1	30.73 ± 12.16	4–48	37.20 ± 24.05	1–76	0.4393
Speaking impairment	8 (53.33)	9 (60.00)	1	17.00 ± 11.56	3–35	14.56 ± 12.01	1–35	0.6760
Interferences with daily activities	7 (46.66)	8 (53.33%)	1	15.00 ± 8.54	3–26	14.38 ± 9.13	1–28	0.8937
Interferences with work	7 (46.66)	8 (53.33%)	1	13.00 ± 11.90	4–27	12.00 ± 11.87	2–32	0.8885

*n*—number of split-mouth defects, VAS units—visual analogue scale units (0 = no discomfort and 100 = unbearable discomfort), SD—standard deviation.

## Data Availability

The data presented in this study are available on request from the corresponding author.

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
