# Peer review of "Efficacy of Guided Tissue Regeneration Using Frozen Radiation-Sterilized Allogenic Bone Graft as Bone Replacement Graft Compared with Deproteinized Bovine Bone Mineral in the Treatment of Periodontal Intra-Bony Defects: Randomized Controlled Trial"

_jcm, 2023, doi:10.3390/jcm12041396_

Round 1

Reviewer 1 Report

The authors performed a randomized controlled trial comparing the efficacy of frozen radiation-sterilized allogenic bone graft as bone replacement graft compared to deproteinized bovine bone mineral for guided tissue regeneration of periodontal intrabony defects.

The manuscript is well organized, the methods are clearly descirbed and the results are completely reported. The RCT is well conceived and performed by the authors.

My judgement regarding the article is absolutely positive.

Author Response

Thank you very much for revision our manuscript.

Reviewer 2 Report

I reviewed your paper investigating grafting materials used for treatment of intrabony defects in periodontitis patients.

The paper was really nicely written, with the methods clearly described, including statistical analyses used.

You provided clear data, with several variables measured. 

A small point, you need to clarify that strictly speaking it is only the px that is blind to the treatment. Unless the different graft materials look identical too each other, the surgeon would surely know the difference?

Also, how did you rule out surgical technique as making a difference? You provide data in Table 3, but it is not clear how you reached those figures.

Thank you

Author Response

Thank you very much for revision our manuscript. 

a) In response, this sentence has been modified in the manuscript to clarify the blinding of patients and those who evaluated the clinical and radiographic results. "Even though the surgeon was not blint to the treatment allocation, patients and researches who evaluated the clinical and radiographic parameters were unaware of the allocation of defects to the tested interventions."

b) The surgical technique for both the test and control bone replacement grafts was identical, and treatments were not allocated until the intrabony defect was exposed and fully debrided. Once the defects were filled with the allocated bone replacement graft a membrane was applied in both test and control groups. The surgical technique, therefore, could not have any impact on the reported results.

On behalf of the  authors we hope these changes allow for the  publication of this manuscript.

Reviewer 3 Report

INTRODUCTION

The null’s hypothesis and alternate hypothesis should be mentioned at the first of the paragraph followed by elucidating the aim of the study which is highlighted.

MATERIALS AND METHODS

Double spacing has been highlighted.

Kindly mention the type of suture material used.

Was there a periodontal dressing given? If yes kindly mention.

STATISTICAL ANALYSIS

Kindly mention the parametric and non parametric tests used for analysis with explanation of why it is applied.

CONCLUSION

Punctuate the highlighted sentence.

If possible kindly add clinical pictures demonstrating the defect fill and procedure.

Overall, the manuscript is well written and requires minor corrections.

Author Response

Thank you very much for  the revision of our manuscript.

As advised by the reviewer the null hypothesis has been introduced as follows: The null hypothesis was based on similar behaviour of both bone grafts in terms of improvement of clinical and radiological parameters."

The requested information has been added to the manuscript ( double space, information on the suture and on the periodontal dressing.)

In the statistical methodology the parametric and non-parametric tests have been identified in the text.

Unfortunately, no clinical pictures were taken.

On behalf of the authors we hope these changes allow for the publication of this manuscript.